# Oxidative Stability and Pasting Properties of High-Moisture Japonica Brown Rice following Different Storage Temperatures and Its Cooked Brown Rice Flavor

**DOI:** 10.3390/foods13030471

**Published:** 2024-02-02

**Authors:** Lingyu Qu, Yan Zhao, Yanfei Li, Haoxin Lv

**Affiliations:** School of Food and Strategic Reserves, Henan University of Technology, Zhengzhou 450000, China; 2021920031@stu.haut.edu.cn (L.Q.); liyanfei@haut.edu.cn (Y.L.); lvhaoxin0129@126.com (H.L.)

**Keywords:** low temperature storage, high-moisture japonica brown rice, oxidation, pasting properties, flavor

## Abstract

The study proposed to investigate the impacts of storage temperatures (15, 20, 25 °C) on the oxidative stability (peroxide value, carbonyl value, malondialdehyde content) and sensory attributes (pasting properties, cooked brown rice flavor) of high-moisture japonica brown rice. According to the findings, the peroxide value, the carbonyl value, and the malondialdehyde content of high-moisture japonica brown rice stored at a temperature of 15 °C exhibited consistently low levels, and the pasting properties were favorable. In addition, 22 out of 51 flavor volatiles were screened as key differential volatile flavor compounds in cooked brown rice via a combination of ANOVA and orthogonal projections to latent structures-discriminant analysis (OPLS-DA). Among them, 3-heptylacrolein had an aroma of fat and mushroom, and its contents were higher at 15 °C and 20 °C. These findings could serve as a valuable reference for storing high-moisture japonica brown rice under low temperature conditions as well as for investigating the flavor characteristics of cooked brown rice derived from this variety.

## 1. Introduction

Rice, as a primary dietary staple, serves as sustenance for almost half of the global populace [1]. Rice is often milled to obtain white and brown varieties. However, white rice is less nutritious due to over-milling [2]. Brown rice is a variety of rice that solely experiences removal of the husk, thereby preserving the embryo, endosperm, and bran [3]. Recent studies have shown that brown rice contains numerous phytonutrients, including polyphenols, dietary fiber, minerals, vitamins, carotenoids, and γ-oryzanols [4]. Therefore, brown rice is getting more and more attention because of its high nutritional value [5].

Brown rice is more susceptible to spoilage than milled rice due to the lack of protection from the rice husk [6]. Furthermore, the storage conditions of brown rice, including temperature and moisture content, significantly impact its overall quality throughout the storage period. For example, Ziegler et al. [7] demonstrated that there was a slight increase in the hardness of brown rice when it was stored at high temperatures (32 °C and 40 °C). The storage moisture content of cooked rice grains was 15.70%, which was found to have the best texture [8]. Similarly, brown rice with 15.5% moisture content had better color and lower fatty acid values when stored at 15 °C [9]. However, there is less information on the oxidative stability and the pasting properties of high-moisture brown rice with 15.5%.

As the demand for high-quality and nutritious food grows among consumers, the fragrance of cooked rice has emerged as a key determinant in assessing its overall excellence [10]. However, the fragrance of rice is dictated by a combination of volatile compounds with odor activity [11]. Several research studies have demonstrated that over 200 volatile compounds contribute to enhancing the delightful flavor of cooked rice. Among the various compounds, there is a diverse array of substances, including ketones, alcohols, aldehydes, and esters [12]. In particular, aldehydes like hexanal, nonanal, decanal, octanal, and methionine are believed to undergo decomposition primarily through lipid oxidation. This degradation process is responsible for a significant portion of the flavor constituents found in cooked rice [13]. In addition, the aroma and flavor of brown rice undergo numerous changes during storage due to temperature and other factors. Recent findings suggested that high-moisture brown rice of the japonica variety at 15 °C storage better retained the volatile flavor substances that are beneficial to it [9]. However, information on cooked rice flavor compounds in the different storage temperatures of high-moisture japonica brown rice is very limited.

The primary purpose of this investigation aimed to comparatively analyze the impact of various storage temperatures on the storage stability of high-moisture japonica brown rice by determining the peroxide value, carbonyl value, malondialdehyde, and pasting characteristics. Meanwhile, the relationship between the storage temperature of high-moisture japonica brown rice and the volatile flavor compounds in its cooked brown rice was investigated using HS-SPME/GC-MS. The results of this study will deepen the comprehension of oxidative stability and rice flavor of high-moisture brown rice in the japonica variety when exposed to cold temperatures and provides worthwhile inspirations for the prospective application of low temperature storage techniques for high-moisture brown rice in the japonica variety.

## 2. Materials and Methods

### 2.1. Materials

The cultivar “Sui-Japonica 18” of brown rice was provided by Yihai Kerry Oils & Grains Industries Co., Ltd. (Jiamusi City, Heilongjiang Province, China), which was harvested in August 2022.

### 2.2. Sample Storage

In this study, high-moisture japonica brown rice was utilized, and the moisture content of the samples was regulated at 15.5 ± 0.2% (Appendix A). The brown rice samples underwent a 90 day storage experiment. They were randomly packed into sealed polyethylene bags to maintain their moisture content and placed in a controlled environment with constant temperature and humidity. The relative humidity was kept at 65%, and the temperatures were kept at 15, 20, and 25 °C, respectively. Samples of brown rice were collected every 15 days (Appendix A). The experiment was conducted using an incubator manufactured by Ningbo Southeast Instrument Co., Ltd. located in Ningbo, China.

### 2.3. Extraction of Japonica Brown Rice Oil

The oil samples of the japonica brown rice were refined following the procedure of Elouafy et al. [14] with appreciable revisions. The samples of japonica brown rice were milled into powders, and the oil from the japonica brown rice was extracted using petroleum ether as the extractant in a Soxhlet extraction apparatus for 12 h. Next, all extracts were pooled into a round-bottomed flask, which was then rotary evaporated in a water bath at 40 °C until the solvent was completely removed. The residue was japonica brown rice oil, which was preserved at 4 °C for subsequent analysis.

### 2.4. Peroxide Value, Carbonyl Value, and Malondialdehyde Content

The peroxide value was determined using the device of Liu et al. [15] with appropriate alterations. A total of 0.1 g japonica brown rice oil was weighed accurately, and we then added 9.8 mL chloroform methanol (7:3) solution, 50 μL ammonium thiocyanate (NHSCN) solution, and 50 μL ferrous chloride solution, respectively. Afterward, it was stored at indoor temperatures for 5 min and measured with a spectrophotometer; the absorbance (A) was 500 nm. The spectrophotometric technique was employed to ascertain the carbonyl value in accordance with China National Standards GB 5009.230-2016 [16].

The malondialdehyde content was assayed based on the description of Niu et al. [17]. Firstly, one gram of japonica brown rice flour and trichloroacetic acid solution (5 mL, 10% *w*/*v*) were mixed and then centrifuged for 10 min (10,000 r/min). Secondly, the supernatant (2 mL) and the trichloroacetic acid solution (2 mL, 0.6% *w*/*v*) were put in a centrifuge tube heating for 10 min. Subsequently, it was cooled to indoor temperatures, and the absorbance was recorded at 450 nm, 532 nm, and 600 nm. The formula for calculating the malondialdehyde content was performed using Equation (1), as follows:Malondialdehyde content (μmol/L) = 6.45 × (A_532_ − A_600_) − 0.56 × A_450_(1)

### 2.5. Pasting Properties

The pasting properties were measured by applying the rapid viscosity analyzer (RVA) instrument (RVA series 4500, Perten Instruments, Waltham, MA, USA) based on the methods of Qu et al. [18]. Briefly, 25.0 ± 0.1 mL of water and 3.00 ± 0.01 g of brown rice flour were transferred to the sample cylinder, and the sample was dispersed by placing a stirrer inside the cylinder and stirring up and down 10 times. The test process was computer-controlled and carried out in accordance with the specified test procedure. The parameters of the pasting properties were recorded on account of the viscosity change curve.

### 2.6. Dissection of Flavor Volatiles

#### 2.6.1. Japonica Brown Rice Cooking

Japonica brown rice cooking was performed by virtue of the approach of Zhang et al. [19]. The japonica brown rice sample (10.0 g) was rinsed three times with water and then soaked in an aluminum box with 16 mL of distilled water for one hour. Thereafter, it was steamed in a rice cooker (SF 40HC682, SUPOR Co., Ltd., Zhengjiang, China) at 500 watts for 40 min, then switched off and braised for 20 min. The total cooking time was two hours, and brown rice samples were kept in the aluminum box during the whole cooking process. Importantly, the moisture content of brown rice samples was 15.5% and followed various storage temperatures for 90 days.

#### 2.6.2. HS-SPME/GC-MS Analytical Procedures

The headspace solid-phase microextraction (HS-SPME) procedures were set up with reference to the system depicted by Li et al. [20]. First, the cooked japonica brown rice prepared according to method 2.6.1 was quickly transferred to a 60 mL brown headspace bottle, which was sealed and placed in a water bath at 70 °C. At the same time, an SPME fiber (DVB/CAR/PDMS, 50/30 µm) was inserted into the cap of the bottle for extracting and absorbing the volatile flavor compounds for 30 min. Subsequently, the SPME fiber containing flavor volatiles was rapidly injected into the GC inlet and resolved using an autosampler at a temperature of 250 °C for a duration of 5 min. In addition, the GC-MS run conditions were set up as previously reported [9].

The flavor volatiles were identified by comparing the normal mass spectral reservoirs from NIST08 and Wiley8. The criterion for volatiles identification necessitates a mass spectrum matching score of 80% or above. The amount of the identified volatiles was computed by dividing the individual peak area by the cumulative area of all peaks.

### 2.7. Statistics

The data underwent analysis using a one-way ANOVA in SPSS 26.0 with a significance level established at *p* < 0.05 in accordance with Duncan’s test. Each experiment was performed three times, and the data obtained were reported as the mean values with the standard error.

Principal component analysis (PCA), fold change (FC), variable importance in project (VIP), and orthogonal partial least squares-discriminate analysis (OPLS-DA) were conducted by the MA website (https://www.metaboanalyst.ca/ (accessed on 17 December 2023)). Origin 2023b was used for cluster analysis and correlation analysis.

## 3. Results and Discussion

### 3.1. Oxidative Stability

Compared to paddy, brown rice is less stable in storage as a result of the absence of the protective husk. However, the pivotal conundrum combined with brown rice storage is rancidity, which is caused by the course of lipid oxidation [21]. Therefore, we evaluated the oxidative stability of japonica brown rice with high water content by measuring the peroxide value, the carbonyl value and the malondialdehyde content during storage.

Since hydroperoxides are important products in the oxidation of fats, the peroxide value is an essential criterion to determine the primary oxidation of fats [22]. According to Figure 1A, the initial peroxide value of unstored brown rice samples was 1.65 meq/kg, but it rose to 5.12, 6.07, and 7.31 meq/kg after 90 days of storage at 15, 20, and 25 °C, respectively. In addition, it was noticed that the peroxide value of high-moisture japonica brown rice showed an increasing trend with increasing storage times at different temperatures, and the highest peroxide value was obtained when stored at 25 °C for 90 days compared to 15 °C and 20 °C. This aligned with the examinations of Liu et al. [23], who stored the brown rice at 15, 25, and 35 °C for 270 days and discovered that its peroxide value increased with storage time, and high temperature conditions resulted in the highest level of peroxide value compared to the conditions at 15 and 25 °C.

The carbonyl value represents the degree of secondary oxidation of the rice bran during souring and deterioration [24]. Consequently, the malondialdehyde can also be used to monitor food freshness [25]. Figure 1B,C shows that the carbonyl value and the malondialdehyde content of brown rice with high moisture content samples increased with storage time, indicating that the cell membrane integrity was gradually compromised, and the quality deteriorated. Specifically, the carbonyl values and the malondialdehyde content of brown rice samples at 15 °C storage for 90 days were significantly lower than those stored at 20 °C and 25 °C. Similarly, Liu et al. [14] discovered that the carbonyl values of brown rice stored at 15 °C for 270 days were significantly lower than those at 25 °C and 35 °C. Additionally, Muhammad et al. [26] found that the elevated temperature led to increased oxidation and malondialdehyde levels in rice bran over the course of storage. Consequently, it could be inferred that japonica brown rice with high moisture content at a temperature of 15 °C storage retarded the growth of its peroxide value, carbonyl value, and malondialdehyde content, which were beneficial to retaining its freshness.

### 3.2. Pasting Properties

Paste characterization is a keen indicator for assessing the quality deterioration of rice [27]. Changes in the pasting properties of brown rice with high moisture content in the japonica variety at various storage temperatures were exhibited on Figure 2. Among them, peak viscosity, minimum viscosity, breakdown, and pasting temperature increased gradually with storage time, while setback and final viscosity increased and then decreased with storage time, and then increased again. Our results were consistent with Zhong et al. [28], they proven that the peak viscosity, minimum viscosity, final viscosity, setback, and pasting temperature of rice that had been naturally aged for approximately one year exhibited a significant increase in comparison to freshly harvested rice. Moreover, Qu et al. [18] also found that peak viscosity, minimum viscosity, breakdown viscosity, final viscosity, setback and pasting temperature increased with storage time in their rice storage experiments. Shu et al. [29] revealed that the peak viscosity, minimum viscosity, and final viscosity of rice exhibited an increase over the storage period, attributed to the formation of the starch-lipid complex and the decrease in amylase activity. In addition, the increase in pasting temperature of brown rice with high moisture content in the japonica variety over the storage period may be caused by the growth of the outer chain of straight-chain starch.

The pasting temperature of japonica brown rice with high moisture content was significantly lowest after 90 days of 15 °C storage compared to 20 °C and 25 °C (Figure 2F). Similarly, Shu et al. [29] detected that the rice’s pasting temperature consistently remained low at a temperature of 15 °C storage. In addition, some studies have shown that rice with lower pasting temperature has better eating quality [30]. As a consequence, high-moisture japonica brown rice had a lower pasting temperature in low temperature storage compared with high temperature storage to maintain its good eating quality.

### 3.3. Flavor Volatiles

#### 3.3.1. Detection and Flavor Characterization of Flavor Volatiles in Cooked Japonica Brown Rice with High Moisture Content

The flavor volatiles presented in cooked brown rice with high moisture content in the japonica variety were analyzed and examined using the HS-SPME/GC-MS technique to acquire their compositions and relative contents. As indicated in Table 1, a collective of 51 flavor volatiles were detected when brown rice with high moisture content in the japonica variety stored at various temperatures for 90 days was cooked, including 10 alcohols, 2 furans, 12 aldehydes, 4 ketones, 19 hydrocarbons, and 4 others.

Alcohols are formed by the oxidation of unsaturated fatty acids or the decomposition of aldehydes [31]. Among the identified alcohols, benzyl alcohol contributed a slightly sweet flavor [32]; 1-octen-3-ol displayed the odors of mushroom, lavender, and rose [33]; 1-octanol (citrus, oil), 1-nonanol (rose, orange), and 1-hexadecanol (rose) also imparted a pleasant flavor to cooked brown rice.

During storage, rice derives its distinctive flavor from aldehydes produced from certain amino acids and unsaturated fatty acids that were oxidized [34]. Additionally, aldehydes have been proposed to be a vital factor in the general flavor of boiled rice. The aldehydes identified in this study include undecanal, dodecanal, tetradecanal, decanal, octanal, nonanal, beta-cyclocitral, (E)-2-octenal, trans-2-nonenal, 3-heptylacrolein, and (2E,4E)-deca-2,4-dienal (Table 1). Undecanal and dodecanal gave cooked brown rice a rose, floral, and citrusy odor. Two possible biosynthetic pathways have been reported for nonanal (rose, citrus) and decanal (citrus, floral) in brown rice. The first one is formed by the autoxidation of hydroperoxides in the raw brown rice grain. The second is formed by the enzymatic degradation of hydroperoxides (lyase and isomerase) during the cooking of brown rice [35].

Furans were the most abundant heterocyclic compounds. The study found that both 2-pentylfuran and 2,3-dihydrobenzofuran were present in all cooked high-moisture japonica brown rice samples. It has been discovered that 2-pentylfuran, which is responsible for the fruity aroma, is produced through the oxidation of linoleic and linolenic acids [36]. Nevertheless, high levels of 2,3-dihydrobenzofuran generated an unpleasant soya flavor [37]. Similarly, Zhong et al. [38] identified 2-pentylfuran in cooked black rice.

Ketones are formed primarily through the oxidation of polyunsaturated fatty acids [39]. Four ketones were detected in all cooked brown rice. However, only three ketones were detected in the 25 °C group, which were geranylacetone, phytone, and (R, S)-5-Ethyl-6-methyl-3E-hepten-2-one (Table 1). Among them, geranylacetone and phytone were ketones found in cooked rice that have the aroma of fresh rose and jasmine, respectively [38].

As given in Table 1, 19 hydrocarbons were identified in different cooked high-moisture japonica brown rice, in which alkanes and olefins were obtained from lipolysis. In our previous study, hydrocarbons in brown rice with high moisture content in the japonica variety were identified after three months of storage. Specifically, dodecane and tridecane were found to produce an unpleasant petrol odor [9]. Additionally, some studies detected nonadecane in aromatic and non-aromatic varieties [40]. Specifically, beta-elemene imparted a sweet flavor to cooked brown rice. Although many alkanes and olefins were detected in cooked brown rice, there is limited information on their impact on the overall flavor of cooked brown rice.

#### 3.3.2. PCA and OPLS-DA Analysis

The total flavor of cooked rice is affected by the contents of flavor volatiles and their interactions [12]. Therefore, the PCA and the OPLS-DA were used to screen the key volatiles impacting the ultimate flavor of the cooked brown rice. Based on the data of the whole volatile flavor substances (Table 1), a dimensionality reduction analysis was performed by PCA. As shown in Figure 3A, the closer the PC accumulation ratio is to one, the more reliable the PCA model is. Concretely, the PC1 and the PC2 accounted for 79.2% and 13.3% of the total square deviations at 15 °C (red dot), 20 °C (green dot), and 25 °C (blue dot). The cumulative variance contribution rate reached 92.5%, which concluded that the selection of the PC1 and PC2 analysis samples have good reliability.

The OPLS-DA model was employed to screen for significant flavor volatiles associated with cooked brown rice under different storage temperature treatments (Table 1). The OPLS-DA score plots are shown in Figure 3B–D. From Figure 3B–D, it can be found that the storage temperature of brown rice with high moisture content in the japonica variety had a notable impact on the flavor volatiles of its cooked japonica brown rice.

#### 3.3.3. Identification of Key Flavor Volatiles in Cooked Japonica Brown Rice with High Moisture Content

For the purpose of finding the most notable disparity of flavor volatiles between various group, Table 2, Table 3 and Table 4 show that the key flavor volatiles (brown rice samples stored at 15, 20, and 25 °C for 90 days) with *p* < 0.05 and VIP > 1 were screened by ANOVA and OPLS-DA analysis. There were 11 significantly different flavor volatiles in the 15 °C and the 20 °C (Table 2) samples. With 15 °C and 25 °C, 16 flavor volatiles were significantly different (Table 3). In addition, only six flavor volatiles had significant differences in 20 °C and 25 °C temperatures (Table 4).

#### 3.3.4. Dissection of Key Flavor Volatiles in Cooked Japonica Brown Rice with High Moisture Content

Table 2, Table 3 and Table 4 show the fold change of the key flavor volatiles between various groups. However, heat maps can visualize data and help us quickly capture the focus of our research (Figure 4A–C). Particularly, the levels of six flavor volatiles were increased in the cooked brown rice when the brown rice with high moisture content in the japonica variety was stored at 15 °C (Figure 4A,B), including (E)-2-octenal, (R, S)-5-Ethyl-6-methyl-3E-hepten-2-one, 2-pentylfuran, 3-heptylacrolein, beta-cyclocitral, and 2-hexyl-1-decanol. Among them, (E)-2-octenal was an important aldehydic volatile compound in cooked rice with cucumber, herbal, and fat odors [38]; 2-pentylfuran might have been a product of high temperature conditions during the rice cooking [41] and had a fruity odor; 2-hexyl-1-decanol and beta-cyclocitral [42] had aromas of lilacs and citrus, respectively. Noticeably, 3-heptylacrolein (fat, mushroom) had higher levels in the cooked brown rice when the brown rice with high moisture content in the japonica variety was stored at 20 °C (Figure 4C).

The Figure 4A describes that the level of decanal in the cooked brown rice enhanced at 15 °C, and this compound is a key discriminator between fragrant japonica rice and non-fragrant japonica rice in China [43]. The content of 2-Undecenal (orange) was also enhanced in the 15 °C group. It has been reported that nonanal is known for inducing citrus, green, and fatty aromas [44], while benzaldehyde is associated with bitter, almond-like, and fruity aromas [42]. Besides, the formation of fruity, apply, and rose aromas originates from geranylacetone. In Figure 4B, the contents of nonanal, benzal alcohol, and geranylacetone increased in the cooked brown rice after 15 °C storage. However, 2,3-dihydrobenzofuran generated an unpleasant soybean odor at high concentrations, which had high levels in the 25 °C group (shown in Figure 4B,C) [37]. Therefore, high levels of (E)-2-octenal, (R, S)-5-Ethyl-6-methyl-3E-hepten-2-one, 2-pentylfuran, 3-heptylacrolein, beta-cyclocitral, and 2-hexyl-1-decanol and low levels of 2,3-dihydrobenzofuran had a beneficial influence on the flavor of cooked brown rice made from brown rice with high moisture content in the japonica variety at low temperature storage.

### 3.4. Correlation Analysis of Storage Temperatures, Oxidative Parameters, Pasting Properties, and Key Flavor Volatiles

The correlation thermograms between storage temperatures, oxidative parameters, and pasting properties of high-moisture japonica rice and six key flavor volatiles of cooked brown rice are exhibited in Figure 5. In particular, storage temperatures demonstrated a significant positive correlation with peroxide value, minimum viscosity, breakdown, final viscosity, and pasting temperature. This implied that the peroxide value and pasting characteristic parameters of brown rice increase with increasing storage temperature. Analogous estimates were obtained by Liu et al. [23], where brown rice had the significantly highest peroxide value when stored at 35 °C for 225 days compared with 15 °C and 25 °C. Furthermore, Shu et al. [29] found that the values of minimum viscosity and final viscosity in paddy rice increased with increasing storage temperatures. Paraginski et al. [45] found that maize at 35 °C storage for 365 days had the significantly highest pasting temperatures compared to 5, 15, and 25 °C.

## 4. Conclusions

The research explored the influences of storage temperature on the oxidation stability and the pasting properties of high-moisture japonica brown rice as well as the flavor volatiles of its cooked brown rice. The peroxide value, the carbonyl value, and the malondialdehyde content of brown rice with high moisture content in the japonica variety gradually increased with storage time, and changes in these oxidative parameters could be delayed at 15 °C. Additionally, brown rice with high moisture content in the japonica variety had lower peak viscosity, minimum viscosity, breakdown, final viscosity, setback, and pasting temperature in low temperature storage compared with high temperature storage to maintain its good eating quality. Furthermore, the levels of six key flavor volatiles were increased in the cooked brown rice when the brown rice with high moisture content in the japonica variety was stored at 15 °C for three months, including (E)-2-octenal, (R, S)-5-Ethyl-6-methyl-3E-hepten-2-one, 2-pentylfuran, 3-heptylacrolein, beta-cyclocitral, and 2-hexyl-1-decanol. Therefore, our results provided important insights into the impacts of low temperature storage on the oxidative stability, the eating characteristics, and the flavor characterization of high-moisture japonica brown rice.

## Figures and Tables

**Figure 1 foods-13-00471-f001:**
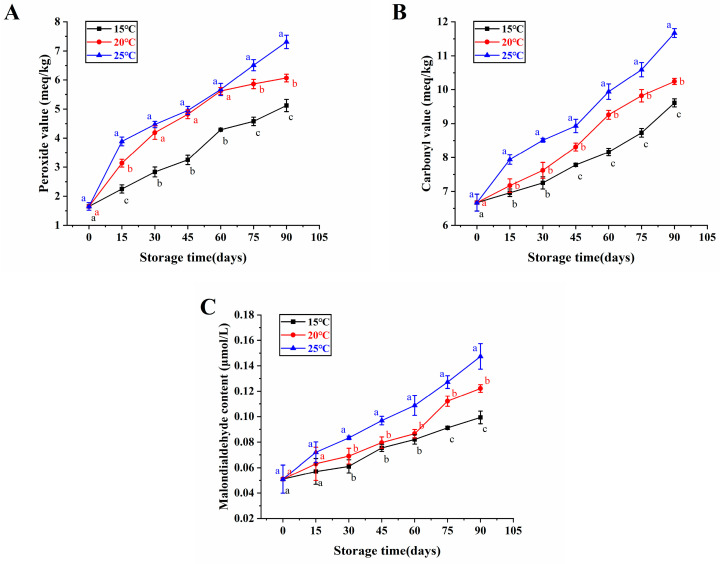
The oxidative stability of brown rice with high moisture content in the japonica variety under various storage temperatures. Peroxide value (**A**), carbonyl value (**B**), and malondialdehyde content (**C**). Distinct letters denote statistically significant variances among various storage temperatures at the same storage duration (*p* < 0.05).

**Figure 2 foods-13-00471-f002:**
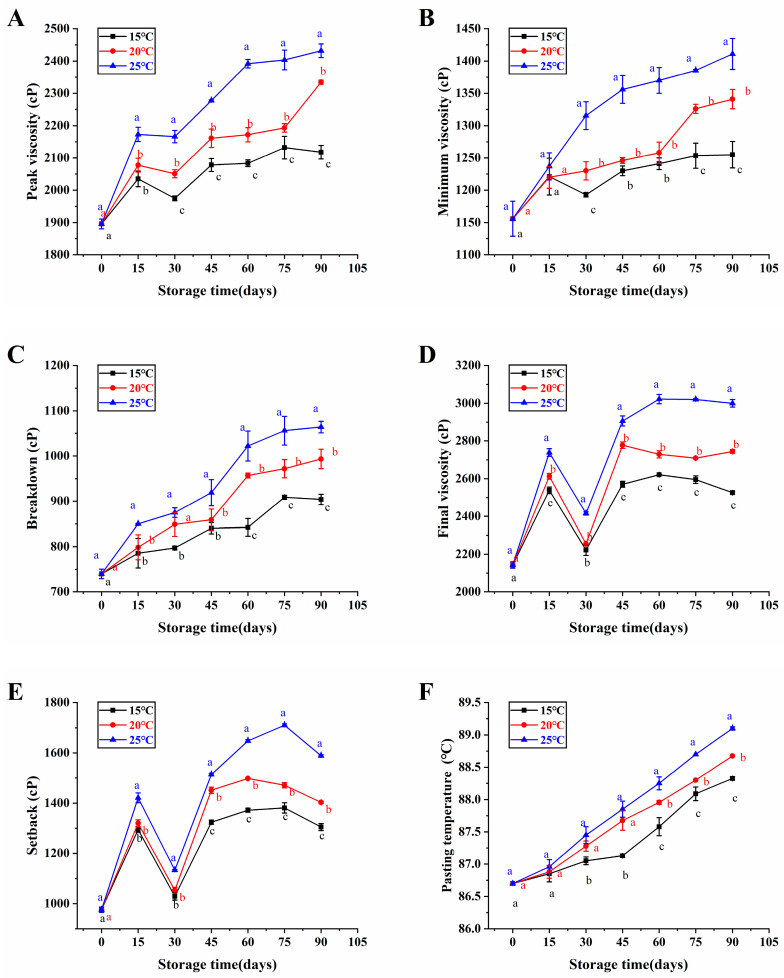
The pasting properties of brown rice with high moisture content in the japonica variety under various storage temperatures. Peak viscosity (**A**), minimum viscosity (**B**), breakdown (**C**), final viscosity (**D**), setback (**E**), and pasting temperature (**F**). Distinct letters denoted statistically significant variances among various storage temperatures at the same storage duration (*p* < 0.05).

**Figure 3 foods-13-00471-f003:**
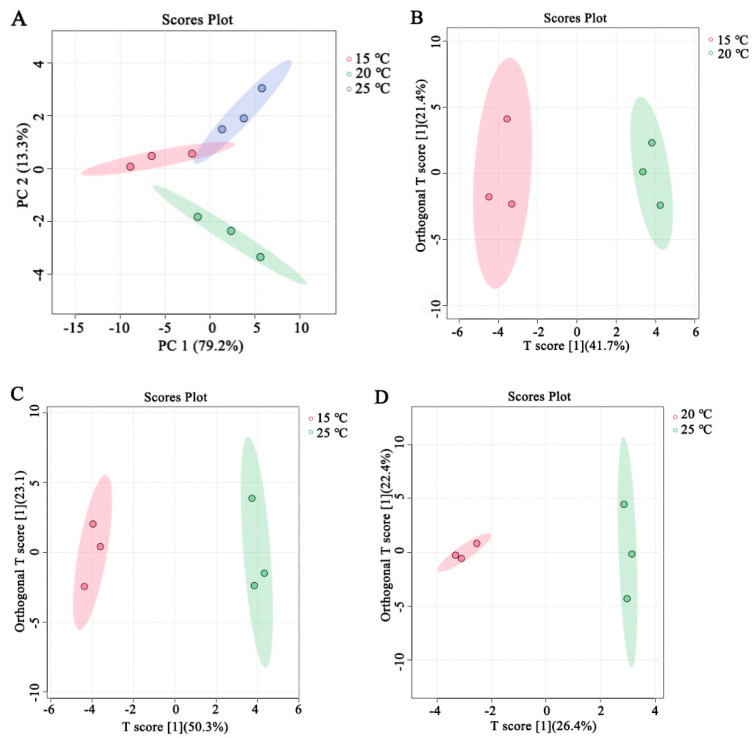
PCA score plots depict the relationship between various storage temperatures of japonica brown rice with high moisture content and flavor volatiles of cooked brown rice (**A**). OPLS−DA of flavor volatiles from cooked brown rice in 15 °C and 20 °C (**B**), 15 °C and 25 °C (**C**), and 20 °C and 25 °C (**D**).

**Figure 4 foods-13-00471-f004:**
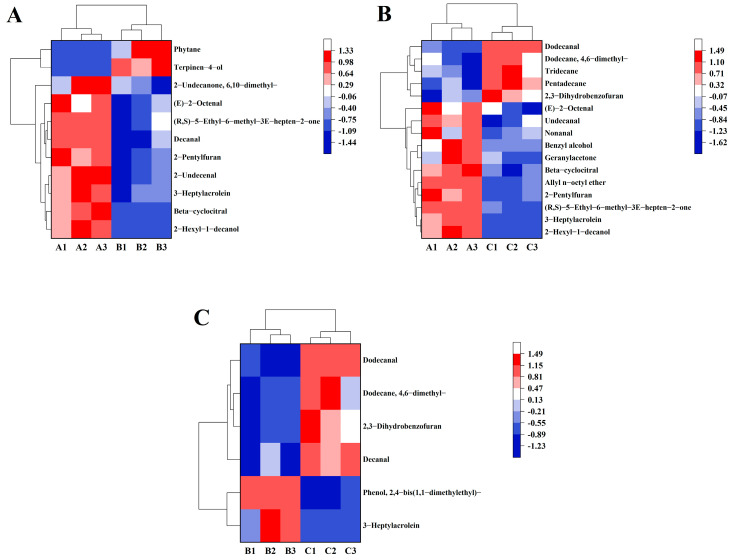
The heat map of the key flavor volatiles in 15 °C and 20 °C (**A**), 15 °C and 25 °C (**B**), and 20 °C and 25 °C (**C**). Red blocks and blue blocks indicate high content and low content of flavor volatiles, respectively. Cooked brown rice samples in the 15 °C group were A1, A2, and A3. Cooked brown rice samples in the 20 °C group were B1, B2, and B3. Cooked brown rice samples in the 25 °C group were C1, C2, and C3.

**Figure 5 foods-13-00471-f005:**
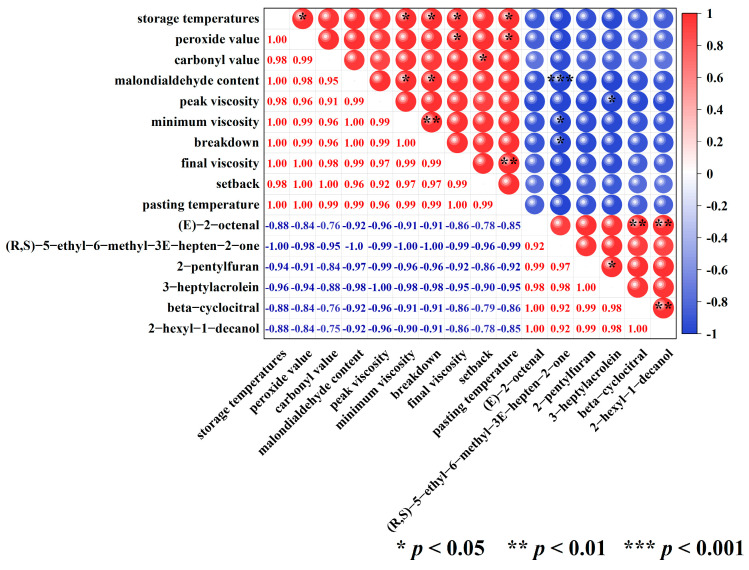
Heat map of correlation among storage temperatures, oxidative indicators, pasting properties, and six key flavor volatiles.

**Table 1 foods-13-00471-t001:** The contents of flavor volatiles (%) in cooked brown rice with high moisture content in the japonica variety using the HS-SPME/GC-MS technique.

No.	Compounds	Storage Temperatures—Storage Time		
		15 °C—90 Days	20 °C—90 Days	25 °C—90 Days
Alcohols				
1	Benzyl alcohol	1.67 ± 0.53 ^a^	0.96 ± 0.35 ^a b^	0.57 ± 0.00 ^b^
2	1-Octanol	1.10 ± 0.08 ^a^	0.89 ± 0.26 ^a^	0.82 ± 0.15 ^a^
3	1-Nonanol	0.79 ± 0.20 ^a^	0.52 ± 0.02 ^b^	0.00 ± 0.00 ^c^
4	1-Dodecanol	0.00 ± 0.00 ^b^	0.53 ± 0.06 ^a^	0.45 ± 0.06 ^a^
5	1-Hexadecanol	0.55 ± 0.23 ^a^	0.57 ± 0.17 ^a^	0.00 ± 0.00 ^b^
6	1-Octen-3-ol	1.71 ± 0.25 ^a^	1.64 ± 0.36 ^a^	1.56 ± 0.42 ^a^
7	2-Hexyl-1-decanol	1.99 ± 0.16 ^a^	0.82 ± 0.06 ^b^	0.79 ± 0.03 ^b^
8	Terpinen-4-ol	1.01 ± 0.07 ^b^	2.58 ± 0.34 ^a^	0.00 ± 0.00 ^c^
9	alpha-Terpineol	0.90 ± 0.31 ^a^	0.70 ± 0.24 ^a^	0.61 ± 0.24 ^a^
10	3,7,11-Trimethyldodecan-1-ol	0.54 ± 0.15 ^a^	0.44 ± 0.04 ^a^	0.35 ± 0.05 ^a^
Furans				
11	2-Pentylfuran	9.88 ± 0.72 ^a^	5.60 ± 0.93 ^b^	4.63 ± 0.67 ^b^
12	2,3-Dihydrobenzofuran	4.78 ± 1.18 ^b^	4.18 ± 0.14 ^b^	7.87 ± 1.33 ^a^
Aldehydes				
13	Undecanal	0.45 ± 0.02 ^a^	0.38 ± 0.12 ^a^	0.35 ± 0.06 ^a^
14	Dodecanal	0.78 ± 0.04 ^c^	0.88 ± 0.06 ^b^	1.23 ± 0.01 ^a^
15	Tetradecanal	0.55 ± 0.04 ^a^	0.52 ± 0.04 ^a^	0.50 ± 0.13 ^a^
16	Decanal	2.43 ± 0.05 ^a^	1.70 ± 0.26 ^b^	2.40 ± 0.07 ^a^
17	Octanal	3.84 ± 0.97 ^a^	2.34 ± 0.77 ^a^	2.32 ± 0.59 ^a^
18	Nonanal	20.94 ± 3.42 ^a^	14.35 ± 3.56 ^b^	14.12 ± 1.84 ^b^
19	2-Undecenal	0.66 ± 0.05 ^a^	0.39 ± 0.06 ^b^	0.00 ± 0.00 ^c^
20	Beta-cyclocitral	0.49 ± 0.06 ^a^	0.26 ± 0.01 ^b^	0.25 ± 0.06 ^b^
21	(E)-2-Octenal	0.73 ± 0.10 ^a^	0.44 ± 0.13 ^b^	0.43 ± 0.14 ^b^
22	trans-2-Nonenal	0.50 ± 0.06 ^a^	0.40 ± 0.04 ^a^	0.41 ± 0.19 ^a^
23	3-heptylacrolein	0.70 ± 0.05 ^a^	0.42 ± 0.06 ^b^	0.32 ± 0.00 ^c^
24	(2E,4E)-Deca-2,4-dienal	0.56 ± 0.06 ^a^	0.55 ± 0.08 ^a^	0.57 ± 0.15 ^a^
Ketones				
25	2-Undecanone, 6,10-dimethyl-	0.45 ± 0.03 ^a^	0.38 ± 0.02 ^b^	0.00 ± 0.00 ^c^
26	Geranylacetone	1.62 ± 0.21 ^a^	1.22 ± 0.33 ^a^	1.21 ± 0.08 ^a^
27	Phytone	1.59 ± 0.24 ^a^	1.44 ± 0.27 ^a^	1.90 ± 1.06 ^a^
28	(R, S)-5-Ethyl-6-methyl-3E-hepten-2-one	0.66 ± 0.02 ^a^	0.45 ± 0.09 ^b^	0.30 ± 0.04 ^c^
Hydrocarbons				
29	Phytane	0.60 ± 0.00 ^a^	0.73 ± 0.07 ^a^	0.81 ± 0.17 ^a^
30	Dodecane	0.84 ± 0.16 ^a^	0.79 ± 0.21 ^a^	0.85 ± 0.11 ^a^
31	Tridecane	0.90 ± 0.10 ^a^	0.87 ± 0.16 ^a^	1.10 ± 0.06 ^a^
32	Tetradecane	2.03 ± 0.21 ^a^	2.23 ± 0.56 ^a^	2.45 ± 0.24 ^a^
33	Pentadecane	0.82 ± 0.15 ^b^	1.02 ± 0.22 ^a b^	1.22 ± 0.10 ^a^
34	Hexadecane	0.69 ± 0.18 ^a^	0.54 ± 0.34 ^a^	0.71 ± 0.27 ^a^
35	Heptadecane	1.19 ± 0.23 ^a^	1.21 ± 0.27 ^a^	1.43 ± 0.16 ^a^
36	Nonadecane	0.57 ± 0.20 ^a^	0.00 ± 0.00 ^b^	0.64 ± 0.20 ^a^
37	Heneicosane	0.00 ± 0.00 ^b^	0.31 ± 0.00 ^a^	0.31 ± 0.04 ^a^
38	Cyclopentane, decyl-	0.27 ± 0.10 ^b^	0.46 ± 0.11 ^a^	0.00 ± 0.00 ^c^
39	Dodecane, 2-methyl-	0.42 ± 0.00 ^a^	0.46 ± 0.06 ^a^	0.51 ± 0.06 ^a^
40	Tetradecane, 2-methyl-	0.31 ± 0.16 ^a^	0.34 ± 0.07 ^a^	0.38 ± 0.14 ^a^
41	Tetradecane, 3-methyl-	0.33 ± 0.05 ^a^	0.43 ± 0.09 ^a^	0.44 ± 0.11 ^a^
42	5-Methylpentadecane	0.28 ± 0.04 ^a^	0.32 ± 0.11 ^a^	0.34 ± 0.20 ^a^
43	1,2-Epoxyhexadecane	0.00 ± 0.00 ^b^	0.60 ± 0.13 ^a^	0.64 ± 0.23 ^a^
44	Dodecane, 4,6-dimethyl-	0.59 ± 0.08 ^b^	0.65 ± 0.01 ^b^	0.76 ± 0.04 ^a^
45	2,6,10-Trimethyldodecane	0.30 ± 0.08 ^a^	0.37 ± 0.02 ^a^	0.40 ± 0.02 ^a^
46	Bate-elemene	0.00 ± 0.00 ^b^	0.45 ± 0.09 ^a^	0.48 ± 0.09 ^a^
47	1,3-Hexadiene, 3-ethyl-2-methyl-	0.37 ± 0.09 ^a^	0.00 ± 0.00 ^b^	0.46 ± 0.19 ^a^
Others				
48	Decane, 1,1’-oxybis-	0.40 ± 0.00 ^a^	0.49 ± 0.21 ^a^	0.00 ± 0.00 ^b^
49	Phenol, 2,4-bis(1,1-dimethylethyl)-	0.00 ± 0.00 ^c^	1.18 ± 0.02 ^a^	0.50 ± 0.08 ^b^
50	Allyl n-octyl ether	1.14 ± 0.01 ^a^	0.00 ± 0.00 ^c^	0.66 ± 0.08 ^b^
51	Tetradecanoic acid	0.00 ± 0.00 ^b^	0.31 ± 0.00 ^a^	0.48 ± 0.19 ^a^

Note: Values with different letters (^a–c^) in a row were significantly different using Duncan’s multiple comparison tests (*p* < 0.05).

**Table 2 foods-13-00471-t002:** The *p*-value, VIP, and fold change of the comparative in 15 °C and 20 °C (15 °C was the experimental group, and 20 °C was the control group).

Compound Name	*p*-Value	VIP	Fold Change
2-Hexyl-1-decanol	0.000	1.518	2.417
Terpinen-4-ol	0.001	1.508	0.390
Decanal	0.009	1.396	1.434
(E)-2-Octenal	0.037	1.288	1.664
2-Undecenal	0.004	1.458	1.670
3-Heptylacrolein	0.003	1.469	1.680
Beta-cyclocitral	0.003	1.504	1.897
2-Undecanone, 6,10-dimethyl-	0.046	1.296	1.165
(R, S)-5-Ethyl-6-methyl-3E-hepten-2-one	0.018	1.336	1.467
2-Pentylfuran	0.003	1.460	0.826
Phytane	0.030	1.319	1.765

**Table 3 foods-13-00471-t003:** The *p*-value, VIP, and fold change of the comparative in 15 °C and 25 °C (15 °C was the experimental group, and 25 °C was the control group).

Compound Name	*p*-Value	VIP	Fold Change
Benzyl alcohol	0.022	1.219	2.930
2-Hexyl-1-decanol	0.000	1.384	2.519
Nonanal	0.038	1.211	1.482
Undecanal	0.050	1.171	1.295
Dodecanal	0.000	1.394	0.633
(E)-2-Octenal	0.042	1.157	1.703
Beta-cyclocitral	0.006	1.319	2.000
3-Heptylacrolein	0.000	1.390	2.188
(R, S)-5-Ethyl-6-methyl-3E-hepten-2-one	0.000	1.388	2.200
Geranylacetone	0.033	1.171	1.336
2-Pentylfuran	0.001	1.388	2.133
2,3-Dihydrobenzofuran	0.039	1.206	0.607
Tridecane	0.042	1.193	0.819
Pentadecane	0.018	1.274	0.675
Dodecane, 4,6-dimethyl-	0.030	1.210	0.776
Allyl n-octyl ether	0.001	1.382	1.719

**Table 4 foods-13-00471-t004:** The *p*-value, VIP, and fold change of the comparative in 20 °C and 25 °C (20 °C was the experimental group, and 25 °C was the control group).

Compound Name	*p*-Value	VIP	Fold Change
Decanal	0.011	1.732	0.707
Dodecanal	0.000	1.906	0.715
3-Heptylacrolein	0.046	1.581	1.302
2,3-Dihydrobenzofuran	0.009	1.801	0.531
Phenol, 2,4-bis(1,1-dimethylethyl)-	0.000	1.919	2.367
Dodecane, 4,6-dimethyl-	0.013	1.765	0.855

## Data Availability

The original contributions presented in the study are included in the article/Appendix A, further inquiries can be directed to the corresponding author.

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
