# Peer review of "Oxidative Stability and Pasting Properties of High-Moisture Japonica Brown Rice following Different Storage Temperatures and Its Cooked Brown Rice Flavor"

_foods, 2024, doi:10.3390/foods13030471_

Round 1
Reviewer 1 Report
Comments and Suggestions for Authors

Comments on the Quality of English Language
Minor improvisation of english required
Reviewer 2 Report
Comments and Suggestions for Authors
The introduction and methods are well written but there are changes need in the results and discussion.
Section 2.2 Please add the number of batches of brown rice samples prepared with 15.5% moisture content for true replication. Also state the number of batches done for each treatment temperature and storage time combination.
Section 2.6.1 You need to add details about how the rice samples were cooked.
Section 3.1 Figure 1 Please add the Duncan's stats letters to the columns in each graph for clarity.
Section 3.2 line 191 By retraction do the authors mean setback?. Please correct. line 198 to 199 You cannot say this since there were still changes with time of storage, it didn't retard aging, it was just lower than the other storage temperatures.
Section 3.3.1 line 225 to 229 Please make a statement about how these oxidation compounds can be negative off flavors.
Table 1 Clarify how long these samples were stored and at what temperature.
Section 3.3.1 line 270 Indicate which flavor compounds are you talking about here.
Figure 3 and Table 2 State the time point of storage for these.
Section 3.3.2 line 276 What do you mean by 'key volatiles'? How do these affect flavor?
Table 2 It is not clear how this table is important to the discussion since you already have the volatiles listed in Table 1. Also how did you do stats for these comparisons, with a t-test? Why did you split them out by two temperature comparisons?
Section 3.3.4 line 291 and 298 These were not upregulated since this is not a metabolic process, they just increased. Please revise. Also, what did they increase compared to? Same question for line 302 and 304 (enhanced) compared to what?
Figure 4 is not useful to my understanding and just duplicates the same data again.
Section 3.4 line 323 What do you mean by thermogram? lines 323 to 334 This paragraph doesn't make any sense, there is no real importance of these correlations. I suggest deleting it. The next paragraph makes more sense.
Conclusion section line 354 "slowed down the increase" Do you mean decreased? Just say decreased as this was not a rate of change. line 357 and 358 How do you define pleasant aroma and better flavor? You cannot say this without a sensory panel.
Please make corrections in the abstract for any changes made in the discussion.
Comments on the Quality of English Language
There a few minor English corrections needed.
Round 2
Reviewer 2 Report
Comments and Suggestions for Authors
Corrections appropriately made.
Comments on the Quality of English Language
You may want to have another look at the grammar.